# Regulation of Plant Responses to Salt Stress

**DOI:** 10.3390/ijms22094609

**Published:** 2021-04-28

**Authors:** Shuangshuang Zhao, Qikun Zhang, Mingyue Liu, Huapeng Zhou, Changle Ma, Pingping Wang

**Affiliations:** 1Shandong Provincial Key Laboratory of Plant Stress, College of Life Sciences, Shandong Normal University, Jinan 250014, China; zhangqikun1016@163.com (Q.Z.); lmy312325@163.com (M.L.); machangle@sdnu.edu.cn (C.M.); 2Key Laboratory of Bio-Resource and Eco-Environment of Ministry of Education, College of Life Sciences, Sichuan University, Chengdu 610064, China; zhouhuapeng@scu.edu.cn

**Keywords:** salt stress, ion transport, osmotic homeostasis, hormone mediation, cell wall regulation

## Abstract

Salt stress is a major environmental stress that affects plant growth and development. Plants are sessile and thus have to develop suitable mechanisms to adapt to high-salt environments. Salt stress increases the intracellular osmotic pressure and can cause the accumulation of sodium to toxic levels. Thus, in response to salt stress signals, plants adapt via various mechanisms, including regulating ion homeostasis, activating the osmotic stress pathway, mediating plant hormone signaling, and regulating cytoskeleton dynamics and the cell wall composition. Unraveling the mechanisms underlying these physiological and biochemical responses to salt stress could provide valuable strategies to improve agricultural crop yields. In this review, we summarize recent developments in our understanding of the regulation of plant salt stress.

## 1. Introduction

The demands on crop yield have risen sharply worldwide to keep up with the rapidly expanding human population over the past twenty years [1]. Thus, how to improve crop yield and quality has become an urgent global agricultural problem. Soil salinization is a major environmental challenge that is threatening agriculture across the world [2]. Approximately 20% of the world’s irrigated agricultural lands are adversely affected by soil salinization [3]. Issues with soil salinization are aggravated by natural environment deterioration, poor irrigation practices, and climate changes [4,5]. Thus, to effectively improve crop yields, it is critical to address the increasingly serious threat of soil salinization.

Two kinds of plants exist: halophytes and glycophytes. Halophytes are salinity-tolerant plants, which have adapted to salinized environments and even benefit from high salt concentrations for optimal growth [6]. In contrast, glycophytes are salinity-sensitive plants, in which growth and development are adversely inhibited by soil salinization [7]. Most crops are glycophytes. High salinity hampers glycophytes’ growth and development, seriously limiting crop productivity and challenging food security. The cultivation and development of salt-tolerant crop varieties are key strategies for increasing crop productivity and yield and ensuring global food security.

Salt stress adversely impacts plants by hindering seed germination, growth and development, and flowering and fruiting [8,9]. The high concentrations of sodium in saline soil limits water uptake and the absorption of nutrients in the plant [10]. Water deficiency and nutritional imbalance induce primary stresses, including osmotic stress and ionic stress. These primary stresses result in oxidative stress and can cause a series of secondary stresses [11]. Together, salt stress leads to various physiological and molecular changes and impedes plant growth by inhibiting photosynthesis, thus reducing the available resources and repressing cell division and expansion [12]. Salt stress affects light-harvesting complex formation and regulates the state transition of photosynthesis [13]. Importantly, the enzyme activities or protein stabilities of the key enzymes in photosynthesis, such as ribulose-1,5-bisphophate carboxylase/oxygenase (RuBisCO), are affected through modulating the glycation under salt stress condition. Salt stress also influences sugar signaling and alters the levels of sugars, such as sucrose, fructose, and glycolysis [14].

As sessile organisms, plants have to develop various strategies to adapt to saline environments. These strategies include a series of signaling transduction pathways that are involved in activities ranging from salt stress sensing to the expression of many salt-stress-responsive genes, which regulate processes including ion transport, osmotic homeostasis, and detoxification. These mechanisms rely on multiple regulatory elements, such as phytohormones, lipids, the cell wall, and the cytoskeleton [10,11,12].

This review briefly describes the recent progress in our understanding of salt stress responses and the underlying regulatory mechanisms in plants, focusing on salt stress signal sensing and transduction. Understanding the molecular mechanisms of plant salt stress regulation will provide insight on how to improve plant salt stress resistance and is a critical step in improving agricultural productivity and food security.

## 2. Salt Stress Sensing

The sensing of salt stress signals initiates a wide array of complex transduction pathways in plants. Early signals that trigger a salt stress response include excess Na^+^, the alteration of intracellular Ca^2+^ levels, and the accumulation of reactive oxygen species (ROS) [4]. Under salt stress, excess Na^+^ is perceived rapidly and triggers downstream sodium stress responses [10] (Figure 1). Salt stress induces ion and osmotic stress, which leads to the elevation of Ca^2+^ in the cytosol; thus, salt stress and changes in osmotic pressure are always associated with the activation of Ca^2+^ channels. Ca^2+^ functions as an important secondary messenger by binding to and activating Ca^2+^ sensors, which evoke a specific calcium signal cascade. The plasma membrane calcium-permeable channel OSCA1 was identified as a putative osmosensor that is required for osmotic stress-induced Ca^2+^ signaling [15,16]. Under osmotic stress, the loss-of-function mutant *osca1* displays impaired Ca^2+^ signal enhancement. The plastidial K^+^ exchange antiporters KEA1/2 and KEA3 also act as an osmosensory component that participates in osmotic stress-induced Ca^2+^ elevation [17]. The *Arabidopsis* monocation-induced Ca^2+^ increases 1 (MOCA1) was identified as a Na^+^-gated calcium-permeable channel and participates in ionic stress-induced Ca^2+^ signaling [18]. The *moca1* mutant is hypersensitive to salt stress and lacks Na^+^-evoked Ca^2+^ enhancement. MOCA1 encodes a glucuronosyltransferase and functions in the biosynthesis of glycosyl inositol phosphorylceramide (GIPC). GIPCs are monovalent-cation sensors that sense Na^+^ and regulate salt stress responses through activating MOCA1 to increase the Ca^2+^ influx [18]. The plasma membrane receptor-like kinase FERONIA (FER) was reported to be required for salt-induced Ca^2+^ spikes and waves to maintain cell wall integrity during salt stress [19]. FER interacts with the pectin component of the cell wall and can sense salt-stress-induced cell wall damage. Cyclic nucleotide-gated ion channels (CNGCs) are calcium-permeable channels that are inhibited by cellular calcium concentrations and are regulated by calmodulin (CaM). Together with BAK1, FER regulates calcium signaling by phosphorylating CNGCs [20,21]. Lastly, salt stress triggers the excessive accumulation of ROS, which also plays a key role in activating calcium signaling. The leucine-rich-repeat receptor kinase, hydrogen-peroxide-induced Ca^2+^ increases 1 (HPCA1), is a hydrogen peroxide sensor that is located in the plasma membrane that detects the increase of H_2_O_2_ under stress stimuli [22]. HPCA1 is required for stomatal closure by mediating the H_2_O_2_-triggered influx of Ca^2+^ [22].

## 3. Regulation of Plant Response to Salt Stress

### 3.1. Ion Balance

Under salt stress, high concentrations of the sodium ion, Na^+^, accumulate in plant cells, ultimately to toxic levels, leading to the disruption of ion homeostasis [4,7]. Plants have developed systems to maintain low levels of Na^+^ by removing Na^+^ from the cytoplasm. This is mainly achieved using Na^+^/H^+^ antiporters, which transport Na^+^ in exchange for H^+^ [11]. The plasma-membrane-localized Na^+^/H^+^ antiporters transport Na^+^ to the apoplast, and the vacuole-localized Na^+^/H^+^ antiporters are responsible for maintaining Na^+^ compartmentation in vacuoles. The salt overly sensitive (SOS) regulatory pathway regulates ion homeostasis through modulating Na^+^/H^+^ antiporters activity during salt stress [4] (Figure 2).

After being triggered by cytoplasmic Ca^2+^, the SOS pathway functions to alleviate salt stress by exporting excess Na^+^. The SOS pathway is comprised of the Na^+^/H^+^ antiporter SOS1, the protein kinase SOS2, and two calcium sensors, SOS3 and SCaBP8 (SOS3-like calcium-binding protein 8) [8]. Under salt stress, SOS3/SCaBP8 perceives the increased cytoplasmic calcium signal and transduces it to the downstream serine/threonine protein kinase, SOS2. SOS3/SCaBP8 recruits SOS2 to the plasma membrane and activates it. Subsequently, SOS2 phosphorylates SOS1, which enhances the plant salt tolerance by increasing the Na^+^/H^+^ exchange activity [12]. SOS1 plays a key role in transporting Na^+^ from the cytoplasm to the apoplast. The efflux of Na^+^ is driven by the proton gradient that is generated from the plasma membrane H^+^-ATPase. Under salt stress, SOS3/SCaBP8-SOS2 also regulates the activities of other transporters involved in ion homeostasis. For instance, the K^+^ and Na^+^ transporters, vacuolar Na^+^/H^+^ exchanger (NHX), vacuolar H^+^-ATPases, and pyrophosphatases (PPase) were reported to be regulated by the SOS pathway [10]. In summary, the SOS pathway maintains the Na^+^ homeostasis and transports excess Na^+^ from the cytosol to the apoplast to prevent the accumulation of Na^+^ to toxic levels.

How the SOS pathway is regulated has been thoroughly investigated. The kinase activity of SOS2 is specifically activated by salt stress stimuli. Under normal conditions, several protein factors inhibit the SOS2 activity and the SOS pathway, such as SOS2-like protein kinase 5 (PKS5), the phosphatase ABA (abscisic acid) insensitive 2 (ABI2), 14-3-3, and GIGANTEA (GI). PKS5 inhibits SOS2 kinase activity via phosphorylation at Ser294 of SOS2. The 14-3-3 protein functions as a protein-kinase-interacting partner of SOS2 at phosphorylated Ser294 and inhibits the kinase activity of SOS2. Interestingly, 14-3-3 proteins also function as a negative regulator of PKS5. Salt stress promotes the interaction between 14-3-3 proteins and PKS5, causing the inhibition of SOS2 and H^+^-ATPase activity [23]. During this process, 14-3-3 proteins bind to Ca^2+^ and are directly modulated by the Ca^2+^ signal. Under non-stress conditions, GI also inhibits SOS2. In response to salt stress, both 14-3-3 and GI are degraded, thereby releasing SOS2 to activate the downstream protein kinase cascade [24,25,26]. In addition, geminivirus RER-interacting kinase 1 (GRIK1) activates SOS2 by mediating the phosphorylation of SOS2 on Thr168 [27]. The calcium-dependent membrane-binding protein, ANNEXIN4 (ANN4), interacts with the SOS2/SCaBP8 complex to fine-tune calcium signaling under salt stress [28]. Besides these activating mechanisms, the SOS pathway deactivates the regulatory system. Once the salt stress is removed, BIN2, a glycogen synthase kinase 3 (GSK3)-like kinase that is the central component in brassinosteroid (BR) signaling, phosphorylates SOS2 on the Thr172 residue to inhibit SOS2 activity and promotes plant growth via BES1/BZR1-mediated transcriptional networks [29].

Under salt stress, plants have to modulate the Na^+^/K^+^ homeostasis through maintaining high K^+^/Na^+^ ratio since excessive Na^+^ often leads to K^+^ deficiency [4,12]. The *sos* mutants show impaired Na^+^/K^+^ homeostasis during salt stress. The uptake of K^+^ is inhibited in *SCaBP8* mutants under salt stress [30]. The potassium transporters, along with voltage-gated channel proteins and their regulators, play important roles in mediating K^+^ absorption, release, and transportation at the cellular and whole-plant levels. For instance, the inward-rectifier K^+^ channel *Arabidopsis* K transporter (AKT1) is a major contributor to K^+^ uptake and transport. Salt stress inhibits the activity of AKT1 [31]. The calcineurin B-like (CBL) proteins and CBL-interacting protein kinases (CIPKs) interact with and activate AKT1. CIPK23 phosphorylates and activates AKT1 to increase K^+^ uptake under K^+^-deficient conditions [32,33]. Two CBL proteins, CBL1 and CBL9, activate the phosphorylation of CIPK23 on AKT1 to ensure the activity enhancement of AKT1 [34]. However, CBL10 competes with CIPK23 for binding to AKT1 and negatively modulates AKT1 activity [30]. A protein phosphatase 2C (PP2C) member, AIP1, mediates the dephosphorylation of AKT1 and negatively regulates CIPK23-activated AKT1 [35,36]. Tonoplast-localized K^+^ channel (TPK1) is regulated by salt stress and modulates the cytosolic K^+^ influx during salinity stress [37]. Salt stress triggers calcium-dependent protein kinase (CDPK) to phosphorylate TPK1 and activate the K^+^ influx [38]. High-affinity potassium transporter, HKT1, provides sodium exclusion and the maintenance of high K^+^/Na^+^ in leaves during salinity stress [39]. Previous studies have indicated that the maintenance of a low Na^+^ concentration in leaves is an essential strategy for plants to enhance their salt tolerance [40,41]. HKT1 mediates low-affinity Na^+^ transport and plays a role in the distribution of Na^+^ from the root-to-shoot xylem sap. ZmHKT1 causes leaf Na^+^ exclusion promotion and is identified as a major salt-tolerance quantitative trait locus (QTL) [42]. HKT1 physically interacts with phosphatase PP2C49, which then inhibits the Na^+^ permeability of HKT1 and negatively regulates salt tolerance [43]. The salt stress response is regulated by the circadian clock in plants. Several proteins that maintain the circadian clock play key roles in regulating salt stress tolerance [44,45]. Recently, studies have demonstrated a new molecular link between clock components and salt stress tolerance in rice. Oryza sativa pseudo-response regulator (OsPRR73) is induced by salt and specifically confers salt tolerance by recruiting HDAC10 to transcriptionally repress OsHKT2;1 and, therefore, regulates rice salt tolerance [46]. Membrane compartment-localized aquaporins might also participate in ion balance regulation through controlling root water uptake, leaf water transpiration, stomatal closure, and small molecule transport in response to salt stress. For instance, the overexpression of the wheat aquaporin TdPIP2;1 improves salt stress tolerance through retaining a low Na^+^/K^+^ ratio under high-salt-stress conditions [47].

### 3.2. Osmotic Homeostasis

Under salt stress, ion imbalance and water deficiency in the plant cell cause osmotic stress. This results in multiple transient biophysical changes, such as the reduction in cell turgor pressure, shrinkage of the plasma membrane, and physical alteration of the cell wall [4]. To alleviate osmotic stress, plants rely on osmotic signaling pathways that regulate processes ranging from gene expression and activation of osmolyte biosynthesis enzymes to water transport systems [41]. Osmolytes, such as proline, polyols, and sugars, accumulate under salt stress. These osmolytes participate in the regulation of osmotic pressure by lowering the osmotic potential in the cytosolic compartment. They also act as signaling molecules to induce ABA accumulation, affect related gene expression, and regulate plant growth under salt stress [48]. Protein kinases act as a convergence point of rapid osmoregulation and salt stress signaling [49]. In response to osmotic treatments, the mRNA levels of histidine kinases, MAPKKK, MAPKK, and MAPK are increased, leading to increased osmolyte synthesis and accumulation [50,51]. Numerous studies have suggested the mitogen-activated protein kinases (MAPKs) are involved in ROS homeostasis [52,53]. For example, ZmMPKs are induced by salt stress and activate oxidative stress regulation to confer salt stress tolerance [51,52,53]. The receptor-like kinase, salt intolerance 1, is activated by MPK3 and MPK6 and functions in the salt-stress-induced oxidative stress response [54]. Osmotic and salt stresses both induce a rapid increase in cytosolic Ca^2+^. The copine protein, Ca^2+^-responsive phospholipid-binding BONZAI1 (BON1), is a critical upstream regulator of osmotic stress signaling since it positively regulates calcium signaling [55]. The disruption of BON1 dampens the cytosolic Ca^2+^ signal in response to osmotic stress.

In plants, salt-stress-triggered ion stress and osmotic stress cause a metabolism imbalance and the toxic accumulation of ROS, inducing plant oxidative damage [40]. Under salt stress, ROS are produced in many plant organelles, such as chloroplast, peroxisomes, mitochondria, and the apoplast. Plant cells sense the accumulated ROS and respond using rapid regulatory mechanisms to scavenge ROS and activate a series of downstream adaptive responses [4,11,12]. ROS function as essential signaling molecules at low levels. Thus, strict control mechanisms are used to balance ROS production and scavenging. Under salt stress, several proteins were found to participate in oxidative stress regulation by activating ROS scavengers or mediating the gene expression of ROS-responsive genes [56]. Several studies have shown that the activities of ROS scavenging enzymes and antioxidants are triggered by salt stress stimuli [57]. For example, the ascorbates peroxidase and catalase are activated by salt stress, improving the tolerance to salinity and oxidative stresses [58]. The overexpression of the ascorbate peroxidases, OsAPXa or OsAPXb, enhances the salt tolerance in rice [59]. Similarly, the constitutive expression of *OsGSTU4* (glutathione *S*–transferase) in *Arabidopsis* also increases the tolerance to salt stress [60]. In rice, the MADS–box transcription factor, OsMADS25, is required for salt tolerance because of its role in maintaining ROS homeostasis [61]. Senescence-associated genes (SAGs) are involved in detoxification in response to salt stress stimuli [62]. The loss of function of SAG29 renders plant seedlings insensitive to salt treatment, while the overexpression of SAG29 results in high sensitivity to salt treatment in *Arabidopsis*. Studies show that NADPH oxidase, respiratory burst oxidase homolog gene, RBOH, mediates ROS synthesis and ROS scavenging to modulate plant development and stress responses [63].

### 3.3. Phytohormone Signaling Mediation—ABA Signaling and BR Signaling

To withstand constantly changing stress conditions in the environment, plants have developed phytohormone-mediated stress resistance mechanisms. Phytohormones play a crucial role in the plant response to salt stress by regulating plant growth and development adaptation. Phytohormones make great contributions to salt stress signal perception and defense system mediation. Nine plant hormones have been well characterized and are divided into two groups: growth promotion hormones and stress response hormones [64]. The growth promotion hormones are composed of auxin, gibberellin (GA), cytokinins (CKs), brassinosteroids (BRs), and strigolactones (SLs). Some of the growth promotion hormones can also play a role in stress response, such as SLs and BRs [64]. The stress response hormones contain abscisic acid (ABA), ethylene, salicylic acid (SA), and jasmonic acid (JA). The crosstalk between different phytohormones also is important for the salt stress response.

Among the nine plant phytohormones, ABA is the most important hormone regulating stress responses. ABA functions as an important secondary signaling molecule to activate a kinase cascade and mediate gene expression during the salt stress response (Figure 3). Under stress conditions, ABA synthesis is induced quickly leading to rapid increases in ABA levels [65]. A high level of ABA activates kinase cascades and improves stress recognition and stress defense reactions [66]. Salt stress limits water uptake, leading to cell dehydration and changes in cell turgor, generating osmotic stresses. Under high-salinity conditions, the increase in endogenous ABA levels causes stomatal closure to regulate water balance and osmotic homeostasis [67]. Thus, osmoregulation is an important function of the ABA-mediated plant salt stress response.

ABA is a 15-carbon isoprenoid that is produced from the methylerythritol 4-phosphate (MEP) pathway via the cleavage of carotenoids. Several enzymes play key roles in the regulation of ABA biosynthesis, such as zeaxanthin oxidase (ZEP), 9-cis-epoxycarotenoid (NCED), and short-chain alcohol dehydrogenase (SCAD) [68]. NCEDs catalyze the rate-limiting carotenoid cleavage reaction. *NCED5* plays an essential role in ABA synthesis and is rapidly induced under salt stress conditions in rice [69]. The small peptide CLAVATA3/ESR-RELATED 25 (CLE25) is secreted from the roots and modulates stomatal control via ABA in root-to-shoot long-distance signaling. The root-derived CLE25 is perceived by the BARELY ANY MERISTEM (BAM) receptors, BAM1 and BAM3, and promotes ABA biosynthesis by upregulating the NCED3 expression in *Arabidopsis* [70]. ABA is primarily synthesized in the root system. ABA is first synthesized in plastids in the root tips and then transported to the shoot and leaves. Changes in ABA levels in roots and leaves have been detected in high-salinity conditions. *Arabidopsis ABA1*, which encodes zeathanxin epoxidase, is the key regulator of ABA synthesis and is also induced by salt stress [71]. Under salt stress, ABA-deficient mutants perform poorly and show salt sensitivity.

Salt-stress-induced osmotic stress also activates ABA signaling transduction pathways. The sucrose-nonfermenting-1-related protein kinase 2s (SnRK2s) are the central components in ABA signaling pathways and play critical roles in osmoregulation [72]. The kinase activities of SnRK2.2/3/6 are inhibited by protein phosphatase 2Cs (PP2Cs) in the absence of ABA. Upon osmotic stress, the ABA receptor proteins, including pyrabactin resistance 1 (PYR1), PYR1-like (PYL), and regulatory component of ABA receptors (RCAR), perceive and bind to the accumulated ABA, which subsequently suppresses the phosphatase activity of PP2Cs [66,72]. As a result, in the absence of inhibition by PP2Cs, SnRK2s are quickly activated. After stress removal, PYL is phosphorylated by the target of rapamycin (TOR) kinase and then disassociates from ABA or PP2C, leading to inactivation of the stress response to promote growth recovery [73]. Recent studies suggest that the upstream kinases, namely, the B2, B3, and B4 Raf-like kinases, are quickly activated and are required for the phosphorylation and activation of SnRK2s in response to early osmotic stress [74]. The Raf-like protein kinases and SnRK2s form the protein kinase cascade that is activated during early osmotic regulation in response to salt stress. The phosphatase ABI1 (abscisic acid insensitive 1) and okadaic acid-sensitive phosphatases of the phosphoprotein phosphatase (PPP) family inhibit the kinase activity of salt-stress-activated SnRK2.4 and regulate primary root growth during the salt stress response [75].

ABA levels increase rapidly under salt stress. Subsequently, salt-stress-induced ABA signaling upregulates the expression of many genes via the targeting of ABA-responsive elements (ABREs) in the regulatory regions of their promoter [76]. ABRE-binding protein/ABRE-binding factor (AREB/ABF) transcription factors are master transcription factors that cooperatively regulate the ABRE-mediated transcription of downstream target genes, enhancing salt stress tolerance. SnRK2s phosphorylate and positively control the AREB/ABF transcription factors [77]. Moreover, the SOS pathway coordinates with ABA signaling. The phosphatase ABI2 (abscisic acid insensitive 2) binds to SOS2 and mediates SOS2 inhibition [78].

Brassinosteroids (BRs) are steroidal hormones that mediate various physiological processes, including cell growth and development, flowering and fruiting [79], and plant stress tolerance. Under salt stress, the biosynthesis of BRs is increased to enhance plant stress tolerance by maintaining ion homeostasis and via osmoregulation. Exogenous BR application reduces ROS production, enhances osmotic regulation and ionic homeostasis, induces the expression of stress-responsive genes, and causes translational changes in stress-responsive proteins. In *Malus hupehensis*, exogenous BR application regulates the activity of Na^+^/H^+^ antiporters and NHX and alleviates salt stress. Exogenous applications of BRs reduce cytosolic Na^+^ levels and increase the absorption of K^+^, which is concomitant with higher salt tolerance [80]. Application with 24-epibrassinolide (EBL), a byproduct from the brassinolide biosynthetic pathway, promotes plant growth and development under salt stress [81]. The overexpression of BR-INSENSITIVE 1 (BRI1)-LIKE receptor homolog 3 (BRL3) promotes the accumulation of osmolytes, such as proline and sugars, which play roles in osmoregulation under salt stress [82,83]. The genetic and phenotypic results of BR-related mutants and overexpression transgenic plants indicate that a proper enhancement of BR signaling benefits plants’ defense against salt stress [83]. The tomato BZR homolog gene, SlBZR1, positively regulates salt tolerance in tomatoes and upregulates the expression of multiple stress-related genes [84]. The BR receptor SERK2 significantly enhances grain size and salt resistance in rice. Adverse high salinity conditions induce SERK2 accumulation to enhance early BR signaling on the plasma membrane to defend against the stress [85].

Under salt stress, BR exerts anti-stress effects by interacting with other hormones, such as ABA. ABA inhibits the growth-promoting effects of BR during salt stress (Figure 3). ABA and BRs antagonistically fine-tune plant growth under salt stress. The BR receptor BAK1 regulates SnRK2.6 and modulates stomatal closure [86]. BIN2 indirectly activates ABI5, the key transcription factor in the ABA signaling pathway, by phosphorylating SnRK2.6 [87,88]. Brassinazole-resistant 1 (BZR1) and BRI1-EMS-suppressor 1 (BES1) are transcription factors that have been elucidated largely in the BR signaling pathway. BZR1 and BZR2 directly inhibit ABI5 expression [89]. BR shares transcriptional targets with ABA, suggesting that BR antagonistically acts with ABA to regulate the stress response.

The BR pathway can also crosstalk with the SOS pathway. BRs induce the accumulation of calcium in the cytosol, which in turn activates the SOS pathway to regulate ionic and osmotic stresses [81,90]. A recent study has reported that BIN2 inhibits SOS2 kinase activity and negatively regulates salt stress tolerance as a molecular switch in the transition to robust growth after salt stress [29].

### 3.4. Cytoskeleton Functions

The cytoskeleton plays important roles in a wide variety of cellular processes, including cell shape determination, cell movement, vesicle trafficking, tip growth, and responses to external stress stimuli [91]. The plant cytoskeleton consists of actin filaments (F-actin) and microtubules (MTs), which constantly undergo dynamic changes in architecture. The cytoskeleton has important functions in the plant salt stress response and helps plants to withstand stress conditions through dynamic organizational changes [41]. Cytoskeleton-associated proteins, including MT-associated proteins (MAPs) and actin-binding proteins (ABPs), bind to the cytoskeleton and regulate cytoskeleton organization. Microtubule-associated protein 65-1 (MAP65-1) regulates microtubule stabilization in response to salt stress. Phosphatidic acid (PA) directly binds to MAP65-1 to modulate its microtubule activity [92].

Salt stress triggers changes in the cytoskeleton architecture by modulating dynamic events, such as nucleation and polymerization, severing and depolymerizing, crosslinking/bundling, and growth/shrinkage [93]. During salt stress, the cortical microtubules are first depolymerized and then reorganized. The destabilization of cortical microtubules enhances salt stress tolerances in plants [94,95]. Similarly, actin depolymerization and stabilization are important for plant salt tolerance. The SOS pathway regulates actin dynamics in response to salt stress. Several studies have provided ample pharmacological evidence to link the SOS pathway to cytoskeleton organization [96,97,98]. The *sos* mutants show abnormal responses to microtubule-associated drugs [99]. The addition of the microtubule-disrupting drug, Oryzalin, causes more death in the *sos1* mutant. Actin reorganization in *sos* mutants is also abnormal in response to salt stress. Disruption of the actin filaments with actin-filament-disrupting drugs, latrunculin A and cytochalasin D (CD), increases death in *sos2* seedlings under salt treatment conditions, while the stabilization of actin filaments with the actin filament stabilizing drug, phalloidin, rescues the lethality phenotype [100].

Calcium is a central secondary messenger that plays an important role in plant salt tolerance. Salt stress induces calcium accumulation in the cytosol and triggers calcium signaling transduction. The cytoskeleton is an important upstream and downstream regulator of calcium signaling [101]. Salt-stress-induced depolymerization of the cortical microtubules leads to the release of Ca^2+^ in the cytosol. The subsequent reorganization of microtubules during salt stress regulates the calcium influx to improve the plant’s salt tolerance. Actin dynamics play a role upstream of Ca^2+^ signaling and serve as a signal to induce Ca^2+^ accumulation. Under salt stress, changes in cytoskeleton organization act as a transducer to activate calcium signaling. The actin-related protein2/3 (Arp2/3) complex, as the actin nucleation factor, functions in actin-dynamics-mediated calcium elevation under salt stress [102]. The Arp2/3 complex regulates mitochondrial-dependent Ca^2+^ stimulation via the regulation of the integrity of mitochondria, which is an important organelle for calcium release in response to salt stress. The actin cytoskeleton also plays a role in decoding the downstream calcium signal. The SOS pathway, which is activated by calcium under salt stress, is closely tied to actin dynamics. SOS3, which serves as a calcium sensor in plants, was reported to play a role in regulating actin dynamics under salt stress. The loss of function in SOS3 leads to an abnormal arrangement of actin filaments in response to salt stress, which can be rescued with external calcium application.

Actin dynamics are also important for ROS production under salt stress. Disordered actin organization triggers an accumulation of ROS levels in the *Arabidopsis* root and acts as the initial signal to activate the salt stress response [96].

### 3.5. Cell Wall Regulation

Accumulating evidence has demonstrated that the cell wall plays an indispensable role in the plant response to salt stress [103]. Salt stress inhibits plant growth and development by repressing cell expansion and division. The cell wall is an important factor for determining cell shape and function and is the first layer of defense against salt stress [12]. Salt stress induces water deficiency in plant cells, causing changes in cell turgor pressure. The cell wall provides mechanical strength to withstand these cell turgor changes [104].

It is believed that the cell wall is one of the early sensors of salt stress. The stress signal is perceived by cell wall sensors localized on the plasma membrane, leading to the induction of downstream responses. One of these cell wall sensors is FER, which is a receptor-like kinase with binding activity to RALF (RAPID ALKALINIZATION FACTOR) peptides [105]. FER senses salt-induced cell wall changes and, in return, sends a downstream signal of cell wall integrity damage. Cell wall leucine-rich repeat extensins (LRX) 3/4/5 function together with RALF peptides and FER to regulate plant growth under salt stress through the modulation of cell wall changes [106].

The cell wall is composed of a complex network of polysaccharides, including cellulose, pectin, and lignin. Secondary cell walls consist of cellulose, hemicelluloses, lignin, and other plant biomass and are distributed in the xylem, fibers, and anther cells. The synthesis of cell wall components is regulated by complicated transcriptional mechanisms under salt stress [107]. The NAC domain and Homeobox HD-ZIP ClassIII (HD-ZIPIII) transcription factors are master regulators of secondary cell wall synthesis [108]. The transcription factors MYB46/MYB83 activates the secondary wall biosynthesis through a transcriptional regulatory program [109]. Cell wall synthesis is tightly regulated by phytohormones, particularly ABA. Both ABA synthesis and signaling are involved in secondary cell wall thickening and lignification [110,111]. SnRK2 kinases, namely, SnRK2.2, 2.3, and 2.6, regulate secondary wall biosynthesis by physically interacting with a NAC family transcription factor, namely NAC secondary wall thickening promoting factor 1 (NST1). SnRK2 phosphorylates NST1 at Ser316, which is a site that is required for the transcriptional activation of downstream secondary wall biosynthesis genes [112].

On the other hand, cell wall components, such as cellulose, lignin, and other polysaccharides, have important biological functions in the plant’s response to salt stress [110,113]. Cellulose, the main component of the cell wall, is synthesized by cellulose synthase (CesA) complexes (CSC) at the plasma membrane and tracks along cortical microtubules at a steady pace guided by the protein cellulose synthase interacting (CSI) 1/POM2 [114]. A cellulose synthase–microtubule uncoupling (CMU) protein has been found to affect the function of CSC. CMU is associated with the plasma membrane and interacts with microtubules to regulate cell expansion and development by modulating microtubule displacement [115]. Recent studies have reported that companion of cellulose synthesis (CC) proteins are required for the association of CSCs with microtubules. CC protein1 (CC1) is a MAP that regulates microtubule dynamics to sustain plant growth under salt stress [116]. The mutation of CESA6, CSI1/POM2, or CC1 confers enhanced sensitivity to salt stress in *Arabidopsis* [115,116,117]. Another cell wall component, namely, lignin, also plays roles in the plant’s adaption to saline conditions. Under saline conditions, plant cells adapt to stress by accumulating lignin and thickening the cell wall. As the central component in lignin biosynthesis, caffeoyl-CoA O-methyltransferase 1 (CCoAOMT1) has been reported to play an essential role in the salt stress response [117]. The loss of function of CCoAOMT1 leads to high sensitivity to salt stress. A recent study showed that β-1,4-galactan, a cell wall component, has specific functions in salt hypersensitivity. The synthase GALACTAN SYNTHASE1 (GALS1) catalyzes the biosynthesis of β-1,4-galactan. Salt stress induces the expression of GALS1 and results in the accumulation of β-1,4-galactan levels in plants, which diminishes salt tolerance. This process is transcriptionally regulated. BARLEY B RECOMBINANT/BASIC PENTACYSTEINE transcription factors BPC1/BPC2 repress the expression of GALS1 and positively regulate plant salt tolerance [118].

## 4. Conclusions

Plants must efficiently adjust their growth to adapt to stress conditions. Salt stress is one of the most serious abiotic stresses experienced worldwide. Identifying the salt stress signaling pathway and characterizing the upstream salt stress sensors could guide approaches to mitigate the negative effects of salt stress on crop yields and ultimately improve agricultural development. Salt stress adversely affects plant growth and development, whereas plants have evolved regulatory mechanisms that allow them to adapt to these adverse conditions. For instance, plant growth is inhibited by salt stress due to decreased photosynthesis. However, the plant also actively slows the growth rate in response to salt stress, leading to increased survival. Plant cells undergo large changes to respond and defend against salt stress. For example, salt stress induces ion stress. In turn, plant cells activate ion transporters and channels to reestablish the ion balance. In the ion transport process, the Na^+^ exclusion, K^+^ influx, Ca^2+^ pump, and Na^+^/H^+^ exchange are all important for plant salt tolerance. In addition, strategies for osmotic and oxidative stress alleviations are also utilized in plants under saline conditions. The identification of upstream regulators, the characterization of a high-resolution sensor, transporter, and channel of Na^+^ and K^+^, and the identification of a novel channel and pool of Ca^2+^ will be areas of active future interest.

High throughput and efficient biotechnologies are important for salt-stress-related gene screening. To date, RNA sequencing has proven to be a fast and effective method for studying the molecular regulation of plant salt tolerance. Transcriptome sequencing techniques have been widely used to identify novel genes that are linked to the regulation of the plant salt stress response [119,120]. The development of next-generation sequencing technology has made it easier to screen for salt-tolerance genes [121,122]. The global survey of transcriptome profiles and microRNA levels of plants in response to salt stress using RNA-Seq has provided useful insights into the mechanisms of salt tolerance [123]. These findings also provide a rich resource for breeding salt-tolerant cultivars through biotechnological approaches using salt-related genes.

## Figures and Tables

**Figure 1 ijms-22-04609-f001:**
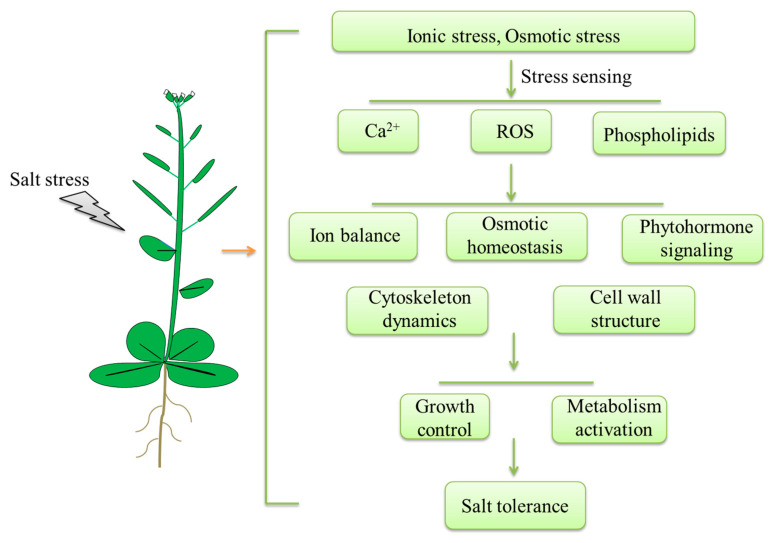
A simplified model of the plant salt stress response. Salt stress primarily causes ionic stress and osmotic stress. After sensing Na^+^ and hyperosmolality, plants accumulate Ca^2+^, activate ROS signaling, and alter their phospholipid composition. These signals activate adaptive processes to alleviate salt stress, including maintaining an ion balance and osmotic homeostasis, inducing phytohormone signaling and regulating cytoskeleton dynamics and the cell wall structure. Subsequently, through an array of signal transduction pathways, plant growth is slowed and metabolism is activated to increase salt tolerance.

**Figure 2 ijms-22-04609-f002:**
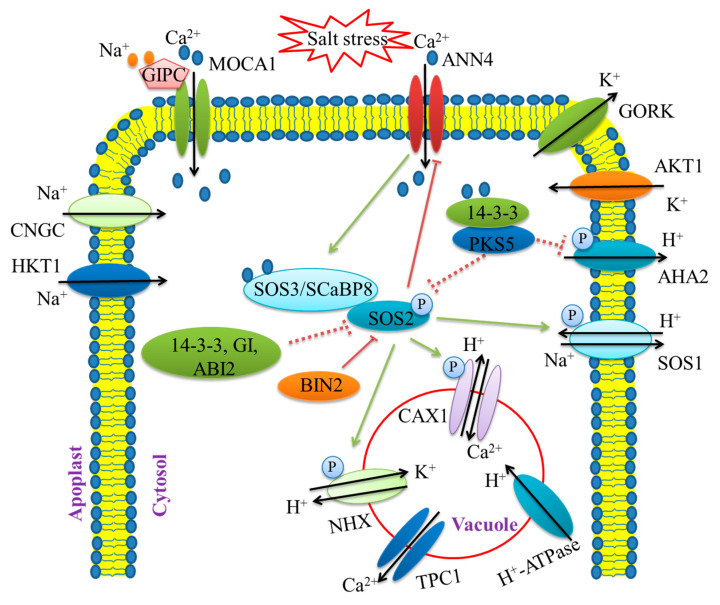
Salt stress triggers ion transport regulation in plant cells. Salt stress induces the accumulation of Na^+^ and Ca^2+^ within the cell. The glucuronosyltransferase monocation-induced Ca^2+^ increases 1 (MOCA1) is as a Ca^2+^-permeable channel in the plasma membrane. Glycosyl inositol phosphorylceramide (GIPC) sphingolipids sense and bind to Na^+^ to activate the MOCA1-mediated Ca^2+^ influx. The cyclic nucleotide-gated ion channel (CNGC) and high-affinity potassium (K^+^) transporter (HKT1) are required for Na^+^ transport into the cell. The inward-rectifier K^+^ channel *Arabidopsis* K^+^ transporter (AKT1) and the outward-rectifier K^+^ channel guard cell outward-rectifier K^+^ channel (GORK) help to maintain the Na^+^/K^+^ balance. The salt overly sensitive (SOS) pathway plays essential roles in Na^+^ exclusion. The calcium sensor, SOS3/SCaBP8, recruits SOS2 to the plasma membrane and promotes SOS2-mediated phosphorylation of the Na^+^/H^+^ antiporter SOS1. Under normal conditions, the kinase SOS2 is repressed by 14-3-3, ABA insensitive 2 (ABI2), and GIGANTEA (GI). Additionally, SOS2-like protein kinase 5 (PKS5) phosphorylates and inhibits SOS2. Under salt stress conditions, 14-3-3 and GI are degraded and release SOS2 to phosphorylate SOS1. Then, 14-3-3 represses PKS5, thereby activating SOS2. Under salt stress conditions, the glycogen synthase kinase 3 (GSK3) kinase, brassinosteroid insensitive 2 (BIN2) fine-tunes the SOS2 activity to prevent overactivation. As a putative Ca^2+^-permeable transporter, the ANNEXIN protein member, ANN4, interacts with SCaBP8 and SOS2 and regulates calcium signaling under salt stress. During salt stress, SOS2 activates the vacuolar H^+^/Ca^2+^ antiporter CAX1 to promote Ca^2+^ enhancement and regulates the vacuolar K^+^/H^+^ exchanger NHX to maintain the K^+^ balance. The arrows and bars indicate positive and negative regulation, respectively. Solid lines and dashed lines indicate direct regulation and indirect regulation, respectively.

**Figure 3 ijms-22-04609-f003:**
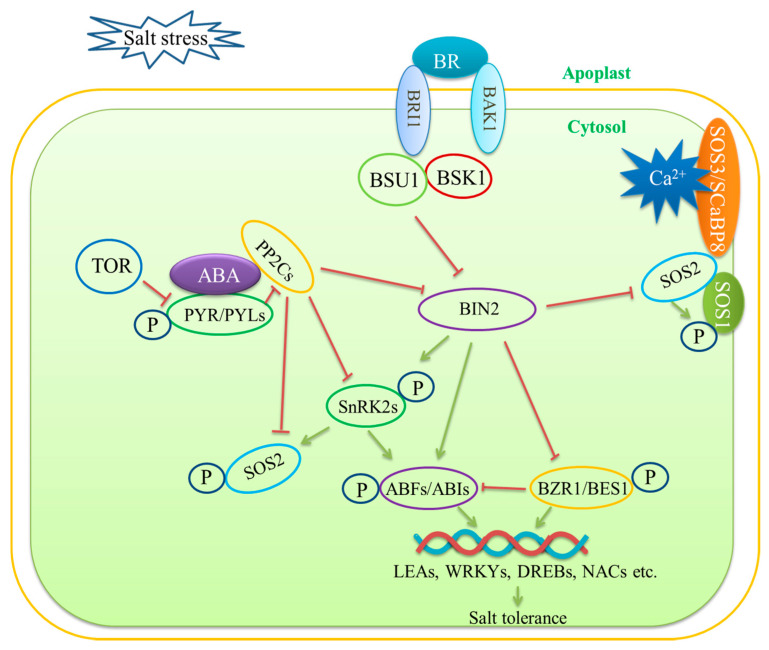
ABA and BR signaling during salt stress. Salt stress promotes abscisic acid (ABA) accumulation. The sucrose nonfermenting-1-related protein kinase2s (SnRK2s) and the clade A type 2C protein phosphatases (PP2Cs) play key roles in mediating the crosstalk between ABA and salt stress signaling. The ABA receptors, PYRABACTIN RESISTANCE/PYR-LIKE (PYR/PYLs) sense ABA and repress PP2Cs, thereby activating the downstream kinase SnRK2. SnRK2s phosphorylate the transcription factors ABSCISIC ACID RESPONSIVE ELEMENT-BINDING FACTORs (ABFs) and ABIs to regulate the expression of stress-responsive genes. The target of rapamycin (TOR) phosphorylates PYL and represses ABA signaling and stress responses. ABI2, a member of the PP2Cs, binds to SOS2 to inhibit its kinase activity, thereby negatively regulating salt tolerance. Additionally, under salt stress, SnRKs phosphorylate SOS2 to activate osmoregulation. Salt stress also upregulates BR biosynthesis. The membrane receptor brassinosteroid insensitive 1 (BRI1) senses BR molecules and acts with its coreceptor BRI1-associated receptor kinase 1 (BAK1) to initiate the downstream phosphorylation cascade. BRI1 and BAK1 transduce the BR signal to BR signaling kinase 1 (BSK1) and activate BRI1 suppressor 1 (BSU1). BSU1 inhibits BIN2 and promotes the transcription factors BZR1/BES1 to induce the expression of BR-responsive genes, which enhances salt tolerance. Under salt stress, BIN2 phosphorylates and inhibits SOS2. This phosphoregulation by BIN2 prevents SOS2 overactivation. Arrows and bars indicate positive and negative regulation, respectively. Solid lines and dashed lines indicate direct regulation and indirect regulation, respectively.

## Data Availability

Not applicable.

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
