# Peer review of "Regulation of Plant Responses to Salt Stress"

_ijms, 2021, doi:10.3390/ijms22094609_

Round 1

Reviewer 1 Report

Dear Sir or Madam,

the manuscript „How Plant Mediate Salt Stress: response and signaling“ comprehensively addresses the mechanisms behind plant tolerance to drought stress. The authors did an excellent job – the review is very informative, well-structured and reports state of the art information. It is definitely needs to be published in IJMS after some minor revision.

Minor remarks:

  1. Line 36: what do you mean as „crop products“? You need to rephrase this sentence, otherwise it is difficult to understand
  2. Lines 43-44: I think, it is too simplified – oxidative stress is a part of the general plant response to stress.
  3. Line 64: may be, “associated”?
  4. Line 95: I don’t really understand this title – ay be “Regulation of plant response to salt stress”?
  5. You are using two terms throughout the text – “salt stress” and “salinity stress”. What is the difference?
  6. Line 190: too many “shrinkage”
  7. Lines 204 and 205, and everywhere as applicable “2+” must be an upper script.
  8. Line 228: remove “S” from “phytohormoneS”
  9. Line 235: I would not say that strigolactones and brassinosteroids are not related to stress. What is about karrikins?
  10. Line 301: why you say here “endogenous”? Drought cannot affect exogenous ABA levels – do I understand it correct?
  11. Line 365: replace “second” with “secondary”
  12. I think, involvement of sugar signaling and, probably, glycation of key regulatory proteins in stress response needs to be highlighted (Int. J. Mol. Sci. 2019, 20, 2366; doi:10.3390/ijms20092366).

Author Response

Reviewer #1 (Comments for the Author): 

Minor remarks:

  1. Line 36: what do you mean as “crop products”? You need to rephrase this sentence, otherwise it is difficult to understand

Response:

As suggested, we rephrased this sentence and replaced “crop products” with “crops” in Line 37.

  1. Line 43-44: I think, it is too simplified - oxidative stress is a part of the general plant response to stress.

Response:

As suggested, we have rewritten the sentence in Line 44-45.

  1. Line 64: may be, “associated”?

Response:

As suggested, we replaced “coupled” with “associated” in Line 70.

  1. Line 95: I don’t really understand this title - may be “Regulation of plant response to salt stress”?

Response:

As suggested, we have changed the title to “Regulation mechanism of plant response to salt stress” in the revised manuscript and made it easy to understand in Line 101.

  1. You are using two terms throughout the text - “salt stress” and “salinity stress”.

Response:

As suggested, we have replaced “salinity stress” with “salt stress” throughout the text to be consistent.

  1. Line 190: too many “shrinkage”

Response:

We have removed the second “shrinkage” in Line 203 in the revised manuscript.

  1. Line 204 and 205, and everywhere as applicable “2+” must be an upper script.

Response:

We have corrected the format of “2+” in Lines 217 and 219.

  1. Line 228: remove “S” from “phytohormonS”

Response:

As suggested, we have replaced “Phytohormones signaling” with “Phytohormone signaling” in Line 241.

  1. Line 235: I would not say that strigolactones and brassinosteroids are not related to stress. What is about karrikins?

Response:

Strigolactones and brassinosteroids also play important roles in regulation of salt stress response. As suggested, we have added the sentence in Line 249-250 to address the important role of strigolactones and brassinosteroids. Because the article space is limited, we selected ABA and BR signaling to summarize in detail.

  1. Line 301: why you say here “endogenous”? Drought cannot affect exogenous ABA levels – do I understand it correct?

Response:

In order to avoid unnecessary confusion, we removed “endogenous” in Line 313. In response to salt stress, ABA levels increase rapidly and activate ABA signaling.

  1. Line 365: replace “second” with “secondary”

Response:

As suggested, we replaced “second” with “secondary” in Line 378.

  1. I think, involvement of sugar signaling and, probably, glycation of key regulatory proteins in stress response needs to be highlighted ( Int. J. Mol. Sci. 2019, 20, 2366; doi: 10.3390/ijms20092366).

Response:

As suggested, we added the involvement of sugar signaling into the revised manuscript in Line 51-52.

Reviewer 2 Report

The paper is focused on plants responses and signalling under exposure to salt stress.

The manuscript is correctly written and organized. However, it needs significant improvement in English style and grammar.

In addition, I recommend adding few references of newly published articles presenting the results of global changes at transcriptomic and microRNA levels in salt-treated plants. Especially, results regarding next generation sequencing (RNA-seq) of salt-exposed plant samples would significantly increase the scientific value of the manuscript.

Author Response

Reviewer #2 (Comments for the Author): 

  1. The manuscript is correctly written and organized. However, it needs significant improvement in English style and grammar.

Response:

The manuscript has been carefully read and revised.

  1. In addition, I recommend adding few references of newly published articles presenting the results of global changes at transcriptomic and microRNA levels in salt-treated plants. Especially, results regarding next generation sequencing (RNA-seq) of salt-exposed plant samples would significantly increase the scientific value of the manuscript.

Response:

Transcriptome sequencing techniques are widely used approaches to increase our understanding of plant salt response regulation. We have added references of newly published articles presenting the results of RNA-seq of salt treated plants in a new paragraph in the Conclusions in Line 462-469.

Reviewer 3 Report

dear Authors, thanks for your manuscript. It fits the purpose of the Journal. I provide suggestions for improving the manuscript.

lines 14-16: you should rewrite ("Beacause...". 

line 47: rewrite

I think that you can also discuss the regulation of photosynthesis and aquaporin

Author Response

Reviewer #3 (Comments for the Author): 
Dear authors, thanks for your manuscript. It fits the purpose of the Journal. I provide suggestions for improving the manuscript.

  1. Lines 14-16: you should rewrite “Beacause...”.

Response:

The abstract has been revised to be more clear and address any grammatical issues in Lines 15-23.

  1. Line 47: rewrite

Response:

As suggested, we have rewritten the sentence in Line 53.

I think that you can also discuss the regulation of photosynthesis and aquaporin

Response:

As suggested, we have discussed the regulation of photosynthesis in Line 48 and aquaporin in Line193-198.